# High donor hemoglobin interacts with pre-transplant recipient neutropenia to modulate mortality after allogeneic hematopoietic stem cell transplantation: An exploratory, single-center, retrospective, real-world study

**Mohammadreza Eslami**[1], **Mahdi Mehrabi**[1]*, **Mehrdad Payandeh**[2]

**1** Student Research Committee, Kermanshah University of Medical Sciences, Kermanshah, Iran,
**2** Department of Internal Medicine, School of Medicine, Kermanshah University of Medical Sciences, Kermanshah, Iran

* mahdi.mhrb76@gmail.com

## Abstract

Prognostication after allogeneic hematopoietic stem cell transplantation remains a critical challenge, and the complex interplay between recipient vulnerability and donor graft characteristics is poorly understood. The primary objective of this retrospective, hypothesis-generating study was to investigate the interaction between pre-transplant recipient neutropenia and donor hemoglobin levels on long-term survival. We performed a pragmatic, single-center, retrospective cohort study on 94 consecutive patients who underwent transplantation at a reference center in Western Iran. Using multivariable survival models, we assessed the independent and interactive effects of pre-transplant factors on 5-year overall survival, with appropriate handling of missing data. The robustness of our central finding was confirmed via sensitivity analyses. Our adjusted multivariable analysis revealed two main findings. First, higher continuous donor hemoglobin was associated with a trend toward increased mortality (Hazard Ratio per 1 g/dL increase = 1.45; 95% Confidence Interval, 0.87–2.39; p = 0.148). Second, the central finding was a statistically significant, qualitative interaction between recipient neutropenia and donor hemoglobin (adjusted HR = 0.44, p for interaction = 0.013). This interaction reversed the potentially deleterious effect of hemoglobin: in the subgroup of neutropenic recipients, higher donor hemoglobin was associated with a protective trend, mitigating the profoundly poor prognosis observed in patients with isolated neutropenia. In conclusion, our study identified a novel and statistically robust interaction, suggesting that the prognostic impact of donor hemoglobin is context-dependent and fundamentally altered by the recipient's baseline immune status. While these preliminary findings provide a compelling rationale for future mechanistic studies, they require urgent validation in larger cohorts and should not be used to guide clinical donor selection.

**Data availability statement:** All relevant data are within the paper and its Supporting information files.

**Funding:** The author(s) received no specific funding for this work.

**Competing interests:** The authors have declared that no competing interests exist.

## Introduction

Allogeneic hematopoietic stem cell transplantation (allo-HSCT) offers a curative potential for many hematological malignances, yet its success is frequently limited by complications such as graft-versus-host disease (GVHD) and disease relapse [1,2]. Improving long-term outcomes requires the identification of robust, pre-transplant prognostic factors to enhance patient selection and tailor therapeutic strategies [3]. While the clinical utility of such factors is clear, the landscape of risk prediction is continually evolving [4,5].

Established prognostic variables include recipient characteristics like age and sex [6,7], pre-transplant laboratory values [8], and derived inflammatory markers such as the neutrophil-to-lymphocyte ratio (NLR) [9]. While the individual effects of such factors are well-documented [10], a deeper understanding of their interplay remains a critical area of investigation. Specifically, severe pre-transplant recipient neutropenia reflects not merely a transient laboratory abnormality, but a heavily pre-treated, depleted bone marrow microenvironment and profound host immunosuppression (the "soil") [10]. Conversely, while routine donor parameters like baseline hemoglobin levels are strictly checked for donor safety, they are rarely investigated as predictors of recipient outcomes [11]. Recent literature has increasingly emphasized that non-HLA donor characteristics, including metabolic state and specific immune subsets mobilized during apheresis, play a profound role in both engraftment kinetics and the modulation of post-transplant complications [12–14]. From a biological standpoint, higher donor hemoglobin within the physiological range may act as a surrogate marker for robust hematopoietic reserve and an inherently vigorous graft (the "seed"). Therefore, it is plausible that the prognostic impact of a recipient's intrinsic vulnerability (e.g., a compromised immune state and a depleted marrow niche) could be modified by the physiological robustness of the donor graft beyond standard HLA matching. Single-center studies from unique real-world settings are valuable for generating new hypotheses in this area, particularly in under-investigated patient populations [15].

Therefore, this single-center, retrospective study was designed as a hypothesis-generating analysis to explore pre-transplant risk factors in our cohort. Our primary objective was to investigate the potential interaction between pre-transplant recipient neutropenia, a marker of host immunocompromise and a hostile microenvironment, and donor hemoglobin, representing graft vigour, levels on 5-year overall survival. Secondary objectives included identifying other independent predictors of outcome in our specific patient population.

## Materials and methods

### Study design and participants

This single-center, retrospective cohort study was conducted and is reported in accordance with the Strengthening the Reporting of Observational Studies in Epidemiology (STROBE) guidelines (S1 Checklist) [16]. We performed a pragmatic analysis of real-world data from the Bone Marrow Transplant Center of Imam Reza Hospital in Kermanshah, Iran, a regional reference center. We identified all consecutive patients

(n = 116) who underwent allo-HSCT for hematological malignancies between September 5, 2015 and March 12, 2024. Patients were excluded for incomplete records with >30% missing data (n = 15) or unavailable survival information (n = 7), resulting in a final cohort of 94 patients for analysis. This threshold was strictly operationalized as missing data across more than 30% of the predefined core baseline and transplant-specific variables, despite rigorous retrospective attempts to retrieve this missing information from electronic and paper-based hospital archives.

### Data collection and definitions

Data were manually extracted from institutional health records by two trained assistants using a standardized form, with high inter-rater reliability (Cohen's $\kappa = 0.92$). The primary outcome was 5-year overall survival (OS), defined as the time from transplantation to death from any cause. Surviving patients were censored on the 28/02/2025. Pre-transplant variables included: 1) recipient and donor demographics (age, sex); 2) clinical features (primary diagnosis, relapse status, HLA-matching); 3) transplant characteristics (conditioning regimen, MNC count); and 4) recipient and donor laboratory data. Laboratory parameters included complete blood counts (defining neutropenia as an absolute neutrophil count <1,500/µL) and comprehensive biochemical panels (defining elevated liver enzymes as ALT/AST >40 U/L).

### Statistical analysis

All statistical analyses were conducted using Stata v17 and R v4.5.1. Baseline characteristics are presented as medians with interquartile range (IQR) or counts with percentages (%). The OS was estimated using the Kaplan-Meier method, with group differences assessed by the log-rank test. For multivariable analysis, we used Gompertz parametric models due to their superior fit (based on AIC/BIC) in our dataset. All primary models were adjusted for recipient age and sex as *a priori* confounders. Given the hypothesis-generating nature of this study, a two-stage significance threshold was employed. For the initial exploratory univariate screening to identify candidate variables for inclusion in the multivariable model, a p-value of < 0.10 was used to minimize the risk of prematurely excluding a potentially important signal. For the final multivariable models and for testing our pre-specified interaction term, a two-sided p-value < 0.05 was considered statistically significant.

Recognizing the modest sample size (n = 94) and number of events (n = 30), we took specific measures to ensure model robustness and prevent overfitting. In accordance with the events-per-variable (EPV) guideline of approximately 10 events per predictor, our primary final multivariable models were restricted to a maximum of three variables.

Crucially, to address potential residual confounding by standard alloHSCT prognostic variables and disease biology (such as primary diagnosis, disease/relapse status at transplant, and conditioning regimen) without violating the EPV constraints, we conducted sensitivity analyses. In these models, key transplant-specific covariates were added sequentially (one at a time) to the primary age- and sex-adjusted Gompertz model to evaluate the stability of the primary interaction effect.

Finally, to confirm that the interaction was not an artifact of the parametric assumptions of the Gompertz model, we repeated the multivariable analysis using a standard semi-parametric Cox proportional hazards model, with the proportional hazards assumption formally verified using Schoenfeld residuals.

Missing data for key covariates (ranging from 5.3% to 18.1%) were handled using multiple imputation by chained equations (MICE) to generate 20 imputed datasets. A sensitivity analysis comparing imputed results with a complete-case analysis was performed to confirm the robustness of our findings.

### Ethical considerations

The study protocol was approved by the Ethics Committee of Kermanshah University of Medical Sciences (approval ID: IR.KUMS.REC.1402.414). This study was a retrospective analysis of data originally collected during routine clinical

care. The requirement for written informed consent was waived by the Ethics Committee given the retrospective nature of the research and the use of de-identified data. The authors accessed the medical records for research purposes from 01/12/2023 to 01/03/2025. During this period, all data were fully de-identified and anonymized upon extraction for analysis.

### AI assistance in manuscript preparation

During the preparation of this manuscript, the authors used the large language model Gemini (Google) for assistance with language editing and to improve the clarity and readability of the text. After using this tool, the authors reviewed and edited the content and take full responsibility for the final version of the manuscript.

## Results

### Cohort derivation and baseline patient characteristics

From an initial 116 consecutive patients assessed for eligibility at our center between September 2015 and March 2024, a final cohort of 94 patients was included for analysis. Twenty-two patients were excluded due to incomplete records with over 30% missing data (n = 15) or unavailable survival information (n = 7). The complete patient selection process is detailed in the STROBE flow diagram (**Fig 1**).

During a median follow-up of 33.6 months (IQR, 4.7–61.7), a total of 30 mortality events (31.9% of the cohort) were recorded. The median overall survival for the entire cohort was not reached. The estimated 1-year, 3-year, and 5-year overall survival rates were 71.3%, 67.3%, and 67.3%, respectively. The baseline demographic, clinical, transplant, and key laboratory characteristics of the study population are summarized in **Table 1**. The cohort had a median age of 43 years (IQR, 31–54), with a majority of male patients (58.5%). Acute Myeloid Leukemia (AML) was the most frequent primary diagnosis (61.7%). The transplant procedures were relatively homogeneous; most patients received a myeloablative

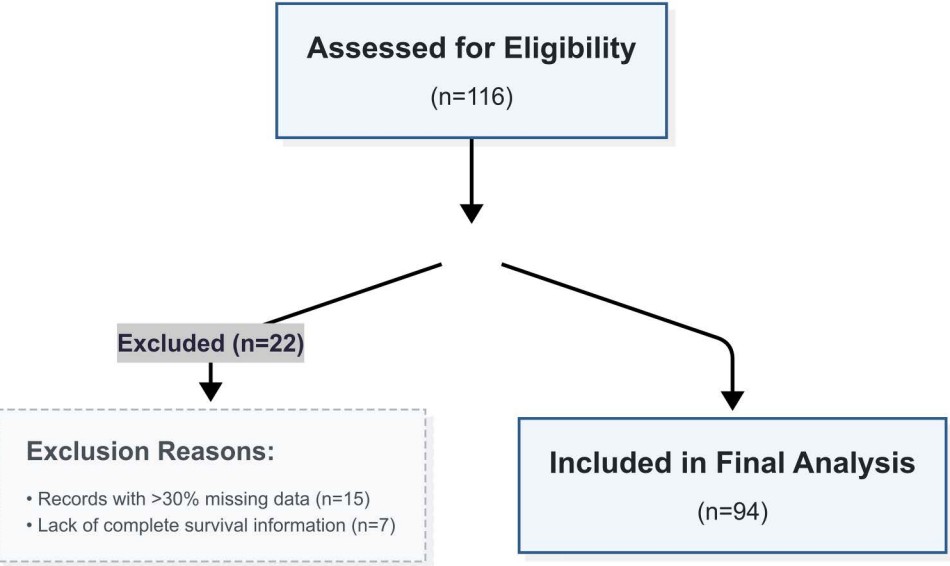

**Fig 1. STROBE flow diagram of patient selection.** Flow diagram illustrating the process of patient inclusion and exclusion. The chart details the initial number of consecutive patients assessed for eligibility, the number of patients excluded, the specific reasons for exclusion with corresponding counts, and the final number of patients included in the definitive analysis.

**Table 1. Baseline characteristics of patients undergoing allogeneic stem cell transplantation (N = 94).**

| Characteristic | Category | Value |
|---|---|---|
| **Recipient Demographics** | | |
| Age (years), Median (IQR) | | 43 (31–55) |
| Gender, n (%) | Male | 55 (58.5) |
| | Female | 39 (41.5) |
| Body Mass Index (BMI, kg/m²), Median (IQR) | (N = 92) | 24.1 (21.2–29.1) |
| **Clinical Characteristics** | | |
| Primary Diagnosis, n (%)[1] | Acute Myeloid Leukemia (AML) | 58 (61.7) |
| | Acute Lymphoblastic Leukemia (ALL) | 14 (14.9) |
| | Aplastic Anemia (AA) | 8 (8.5) |
| | Other | 14 (14.9) |
| Disease Status at Transplant, n (%)[2] | Non-relapsed (in remission/stable) | 82 (87.2) |
| | Relapsed (active disease) | 12 (12.8) |
| **Transplant Characteristics** | | |
| HLA Match Status, n (%)[3] | Full-match | 84 (89.4) |
| | Haploidentical | 10 (10.6) |
| Stem Cell Source, n (%) | Peripheral Blood Stem Cells (PBSC) | 94 (100) |
| ABO Compatibility, n (%) | Matched | 49 (52.1) |
| | Mismatched | 45 (47.9) |
| Conditioning Regimen, n (%) | MAC (Busulfan + Endoxan) | 80 (85.1) |
| | Flu / ATG / Bu | 7 (7.4) |
| | ATG / Bu / Endoxan | 7 (7.4) |
| GVHD Prophylaxis, n (%)[4] | Standard (Cyclosporine-based) | 84 (89.4) |
| | PTCy / ATG / Cyclosporine-based | 10 (10.6) |
| **Key Pre-transplant Laboratory Values** | **Median (IQR)** | |
| Recipient WBC (×10⁹/L) | (N = 93) | 4.4 (1.9–7.4) |
| Recipient Hemoglobin (g/dL) | (N = 93) | 9.3 (8.1–11.2) |
| Recipient Platelets (×10⁹/L) | (N = 92) | 158 (41–283) |
| Recipient ALT (U/L) | (N = 88) | 34 (21–48) |
| Recipient AST (U/L) | (N = 88) | 25 (18–35) |

**Abbreviations:** IQR, Interquartile Range; AML, Acute Myeloid Leukemia; ALL, Acute Lymphoblastic Leukemia; HLA, Human Leukocyte Antigen; MAC, Myeloablative Conditioning; ATG, Anti-thymocyte Globulin; Bu, Busulfan; Flu, Fludarabine; WBC, White Blood Cell Count; ALT, Alanine Aminotransferase; AST, Aspartate Aminotransferase.

[1] Other diagnoses include Myelodysplastic Syndrome (MDS), Chronic Myeloid Leukemia (CML), Myelofibrosis, Non-Hodgkin Lymphoma, Multiple Myeloma, and Congenital Amegakaryocytic Thrombocytopenia (CAMT).

[2] Detailed remission states (e.g., CR1, CR2) were grouped into "Non-relapsed" versus "Relapsed" due to registry data categorization limits.

[3] The vast majority of donors (>80%) were related family members.

[4] Standard GVHD prophylaxis (Cyclosporine) was administered to full-match recipients, while haploidentical recipients received an augmented regimen including ATG, Endoxan (Cyclophosphamide), and Cyclosporine.

conditioning (MAC) regimen (85.1%) and a graft from a fully matched donor (89.4%). A critical pre-transplant vulnerability, neutropenia, was present at the time of assessment in 41 patients (43.6%).

### Initial univariate survival analyses and identification of prognostic signals

To identify potential prognostic factors, we first conducted univariate survival analyses using two complementary methods. The results from these initial screenings are fully presented in **Table 2**.

First, log-rank tests were used to assess survival differences across categorical subgroups. This analysis revealed several trends toward an association with poorer overall survival (OS) at a $p < 0.10$ significance threshold. The most notable associations were observed for pre-transplant recipient neutropenia ($p = 0.052$), a high Neutrophil-to-Lymphocyte Ratio (NLR > 3, $p = 0.093$), and high serum uric acid levels (>7.3 mg/dL, $p = 0.078$). Kaplan-Meier survival curves for these key initial comparisons are illustrated in **Fig 2**.

Second, to further investigate these and other continuous variables, univariate Gompertz proportional hazards regression was performed. This analysis corroborated the log-rank test findings and identified several additional significant predictors of increased mortality. These included higher recipient age (≥50 years; HR 1.87, $p = 0.093$), elevated liver enzymes (ALT and AST, $p < 0.07$ for both), and importantly, higher continuous donor hemoglobin (HR per 1 g/dL increase = 1.017;

**Table 2. Univariate analysis of pre-transplant factors associated with overall survival.**

| Factor | Hazard Ratio (HR) / Stat | 90% Confidence Interval | P-value |
|---|---|---|---|
| **Log-rank Tests** | | | |
| Recipient Leukopenia | $\chi^2(1) = 3.34$ | N/A | 0.068 |
| Recipient Neutropenia | $\chi^2(1) = 3.79$ | N/A | 0.052 |
| Recipient Uric Acid > 7.3 mg/dL | $\chi^2(1) = 3.11$ | N/A | 0.078 |
| Recipient PTH < 15 pg/mL | $\chi^2(1) = 2.88$ | N/A | 0.089 |
| Recipient High NLR | $\chi^2(1) = 2.83$ | N/A | 0.092 |
| **Mantel-Cox** | | | |
| Conditioning: Flu/ATG/Bu vs. MAC | Rate Ratio: 0.368 | 0.12 to 1.093 | 0.061 |
| **Gompertz Regression** | | | |
| Age ≥ 50 years vs. < 50 | 1.871 | 1.014 to 3.454 | 0.093 |
| Neutropenia (Yes vs. No) | 1.92 | 1.108 to 3.834 | 0.055 |
| Uric Acid > 7.3 vs. ≤ 7.3 (mg/dL) | 2.028 | 1.072 to 3.838 | 0.068 |
| ALT (per unit increase) | 1.007 | 1.001 to 1.014 | 0.066 |
| AST (per unit increase) | 1.015 | 1.002 to 1.029 | 0.062 |
| Leukopenia (Normal WBC vs. Leuko.) | 0.506 | 0.271 to 0.944 | 0.072 |
| NLR (Normal vs. High) | 0.525 | 0.280 to 0.985 | 0.092 |
| PTH < 15 vs. ≥ 15 (pg/mL) | 0.379 | 0.156 to 0.919 | 0.072 |
| Donor Hemoglobin (per unit increase) | 0.98 | 0.794 to 1.220 | 0.903 |
| Primary Diagnosis: other malignant vs. leukemia | 1.78 | 0.790 to 4.000 | 0.24 |
| Primary Diagnosis: Non-Malignant/BMF vs. Leukemia | 0.55 | 0.170 to 1.870 | 0.42 |
| Relapse status: relapsed vs. non-relapsed | 0.43 | 0.130 to 1.440 | 0.25 |
| HLA Match: haploidentical vs. fully matched | 0.84 | 0.350 to 2.040 | 0.75 |
| ABO / Blood group mismatch: Yes vs. No | 0.63 | 0.340 to 1.170 | 0.217 |

Abbreviations: HR, Hazard Ratio; CI, Confidence Interval; PTH, Parathyroid Hormone; NLR, Neutrophil-to-Lymphocyte Ratio; ALT, Alanine Aminotransferase; AST, Aspartate Aminotransferase; MAC, Myeloablative Conditioning (Busulfan+Endoxan); ATG, Anti-thymocyte Globulin; Bu, Busulfan; Flu, Fludarabine; Leuko., Leukopenic.

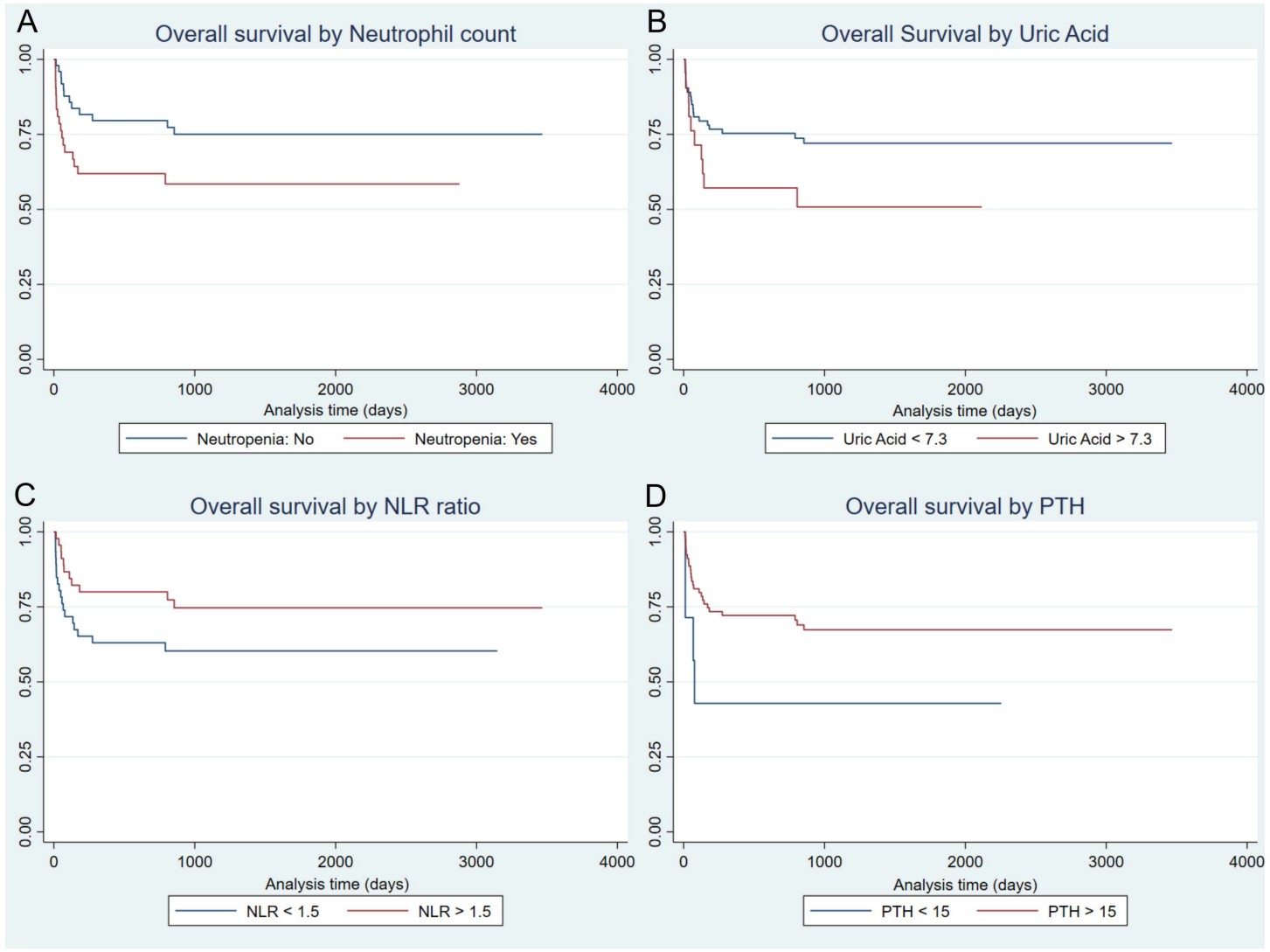

**Fig 2. Kaplan-Meier curves for overall survival based on initial univariable analysis.** Survival estimates for patient subgroups stratified by pre-transplant variables identified in initial univariable screening. Panels display comparisons for **(A)** Recipient Neutropenia status, **(B)** Recipient Neutrophil-to-Lymphocyte Ratio (NLR), categorized at a cutoff of >3, and **(C)** Recipient Uric Acid level, categorized at a cutoff of >7.3 mg/dL. P-values were calculated using the log-rank test. *Abbreviation: NLR, Neutrophil-to-Lymphocyte Ratio.*

90% CI, 1.003–1.031; p = 0.045). Conversely, factors trending toward improved survival included a normal recipient white blood cell count and a normal NLR.

## Independent predictors of mortality in multivariable analysis

To disentangle the independent effects of these correlated factors, we constructed our primary multivariable Gompertz proportional hazards model, adjusting for recipient age and sex. The results, including adjusted hazard ratios (aHRs) and 95% confidence intervals, are presented in the forest plot in **Fig 3**.

Several factors remained statistically significant predictors of mortality. Elevated recipient liver enzymes (ALT and AST) retained their strong association with poorer survival. Subsequently, to evaluate our core hypothesis, we constructed our

| Multivariate Analysis of Factors | HR | 95% CI | P-Value |
|---|---|---|---|
| **Increased Mortality Risk** | | | |
| Recipient AST (per unit increase) | 1.019 | 1.004 – 1.035 | 0.015 |
| Recipient ALT (per unit increase) | 1.010 | 1.002 – 1.018 | 0.015 |
| Recipient Neutropenia (Yes vs. No) | 2.135 | 1.019 – 4.477 | 0.045 |
| **Trend Towards Improved Survival** | | | |
| Normal WBC (vs. Leukopenia) | 0.492 | 0.234 – 1.036 | 0.062 |
| Recipient Uric Acid (per unit, MI) | 0.768 | 0.577 – 1.021 | 0.070 |
| **Other Factors** | | | |
| Donor Hemoglobin (per unit increase) | 0.955 | 0.723 – 1.262 | 0.746 |

Adjusted Hazard Ratio (aHR)
0.1    1    2    4    7

**Fig 3. Forest plot of adjusted hazard ratios for overall survival.** The plot displays the results from the final multivariable Cox proportional hazards model. Circles represent the adjusted Hazard Ratios (aHRs) for the primary pre-transplant variables, and horizontal lines indicate the 90% confidence intervals (CIs). All hazard ratios are adjusted for recipient age and gender. An aHR greater than 1.0 indicates an increased risk of mortality. *Abbreviations: aHR, adjusted Hazard Ratio; CI, Confidence Interval.*

primary interaction model. Within this specific model, recipient neutropenia also remained a powerful independent predictor of adverse outcomes (p = 0.045). The main effect of continuous donor hemoglobin did not reach statistical significance (aHR = 1.45; 95% CI, 0.88–2.39; p = 0.148).

### The central finding: A qualitative host-graft interaction

The pre-specified primary analytical goal was to test for an interaction between recipient neutropenia and continuous donor hemoglobin. Our analysis revealed a statistically significant and clinically meaningful negative interaction term (aHR for interaction = 0.44; 95% CI, 0.23–0.84; p = 0.013). This indicates a qualitative or "crossover" interaction, signifying that the prognostic impact of donor hemoglobin is not uniform but is fundamentally altered by the recipient's baseline immune state.

This complex relationship is powerfully visualized by the model-predicted survival curves presented in Fig 4. The plot clearly illustrates that patients with isolated neutropenia (Neutropenia, Low HGB; curve D) experience the poorest predicted survival. In contrast, the addition of a high-hemoglobin graft to neutropenic recipients (Neutropenia, High HGB; curve C) results in a dramatic mitigation of this risk, elevating their predicted survival substantially and demonstrating a clear "rescue effect." This confirms that the prognostic meaning of donor hemoglobin is context-dependent.

### Engraftment kinetics

The median time to engraftment in the study cohort was 10 days (interquartile range: 9–11 days; mean: 10.62 ± 2.60 days). Neither pre-transplant recipient neutropenia nor baseline donor hemoglobin levels significantly affected the time to engraftment. Furthermore, the interaction model incorporating both recipient neutropenia and donor hemoglobin demonstrated no significant impact on engraftment kinetics (p = 0.812).

## Interaction of Neutropenia and Donor HGB on Predicted Survival

Based on a multivariable Gompertz proportional hazards model

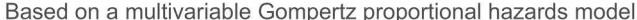

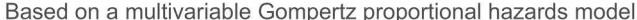

**Fig 4. Model-predicted survival curves illustrating the interaction between recipient neutropenia and donor hemoglobin.** The plot displays adjusted survival probabilities derived from the final multivariable Gompertz proportional hazards model, adjusted for recipient age and sex. Patients are stratified into four mutually exclusive subgroups based on pre-transplant neutropenia status (present/absent) and donor hemoglobin level (Low vs. High, categorized by the cohort median of 13.5 g/dL). The curves visually demonstrate the qualitative interaction: while high donor HGB is associated with worse survival in non-neutropenic recipients (curve B vs. A), it is associated with a marked improvement in survival in neutropenic recipients (curve C vs. D), visually suggestive of a potential 'rescue effect' that requires formal validation. Confidence intervals (95%) for the predicted survival are shown as shaded areas. The table below the plot indicates the number of patients at risk in each subgroup over time. *Abbreviation: HGB, Hemoglobin.*

### Model robustness and confirmatory analyses

To ensure the statistical validity of our central finding, a series of pre-planned confirmatory and sensitivity analyses were performed.

First, to confirm that the interaction was not an artifact of the parametric assumptions of the Gompertz model, we repeated the multivariable analysis using a standard semi-parametric Cox proportional hazards model. The result was highly consistent, again yielding a statistically significant interaction of similar magnitude and direction (p for

interaction = 0.030). Second, to confirm the robustness of findings derived from our multiple imputation dataset, we compared the results for key continuous variables with those from a complete-case analysis. As shown in **S1 Table**, the protective association of lower uric acid was highly consistent across both methods (MI aHR = 0.768 vs. CCA aHR = 0.759), enhancing confidence in the primary analysis.

Third, to address potential residual confounding by key transplant-specific covariates, we conducted targeted sensitivity analyses. Respecting EPV constraints, we sequentially and individually added primary diagnosis group, conditioning regimen, and relapse status to our primary adjusted Gompertz model. In each of these separate sensitivity models, the core interaction between recipient neutropenia and donor hemoglobin remained robust and its significance was not materially altered. Furthermore, none of these added clinical covariates emerged as significant independent predictors of mortality when adjusting for the primary exposures.Finally, exploratory subgroup analyses provided further insights into potential effect modification for other variables, as detailed in **Fig 5A** and **5B**. These analyses suggested, for instance, that the adverse effect of neutropenia was most pronounced in older patients (aged ≥50 years). Notably, sex-based stratification revealed an intriguing divergence regarding donor hemoglobin: while higher donor hemoglobin showed a trend towards increased mortality risk in male recipients (aHR = 1.31; 95% CI, 0.86–2.00; p = 0.204), it demonstrated a potentially protective effect against mortality in female recipients (aHR = 0.69; 95% CI, 0.45–1.06; p = 0.092).

## Discussion

In this single-center, hypothesis-generating study of 94 allo-HSCT recipients, we identified a novel, statistically robust interaction between pre-transplant recipient neutropenia and donor hemoglobin. Our primary multivariable analysis confirmed two key findings: first, that the main effect of higher continuous donor hemoglobin was, paradoxically, associated with a trend toward increased mortality in the overall cohort (aHR 1.45, p = 0.148). Second, and more importantly, a significant qualitative interaction (aHR 0.44, p = 0.013) revealed that this association was fundamentally context-dependent. The deleterious effect of high donor hemoglobin was reversed in neutropenic recipients, in whom it was instead associated with a strong trend toward a protective "rescue effect." In separate adjusted models, established factors, such as elevated pre-transplant liver enzymes, were also confirmed as independent predictors of poorer survival in our cohort.

The central objective of this exploratory analysis was to move beyond linear risk paradigms and investigate the interplay between host vulnerabilities and graft characteristics. Our findings directly address this objective by demonstrating that the prognostic meaning of a key donor characteristic is not fixed but is profoundly altered by the recipient's pre-transplant immune context.

The observed interaction suggests that for a neutropenic host, where poor hematopoietic reconstitution is a primary threat [17], a "high-vigor" graft—for which high donor hemoglobin may be a proxy—could be uniquely beneficial. This "graft-rescues-host" framework, starkly visualized by the model-predicted survival curves in **Fig 4**, reframes the clinical challenge from a simple additive model of risk factors to a more dynamic interplay where the optimal graft is context-specific. This concept of dual, context-dependent roles is a recognized principle in transplant immunology, where alloimmune responses can mediate both detrimental GvHD and beneficial GvL effects [18].

Our observation that higher donor hemoglobin is potentially associated with increased mortality in the absence of neutropenia is a provocative finding that runs counter to clinical intuition and lacks direct precedent in donor selection literature, which has primarily focused on canonical factors like age and sex [17]. However, several plausible, non-mutually exclusive hypotheses can be proposed to explain this main effect. First, high hemoglobin may not be a direct causal agent but rather a surrogate for an unmeasured systemic pro-inflammatory state in the donor. This is conceptually analogous to the situation in the recipient, where pre-transplant inflammation, often measured by C-reactive protein (CRP), is a powerful and established independent predictor of increased NRM and poorer OS [19,20]. It is therefore conceivable that a state of "stress erythropoiesis" in the donor, reflected by high hemoglobin [21], marks the transfer of

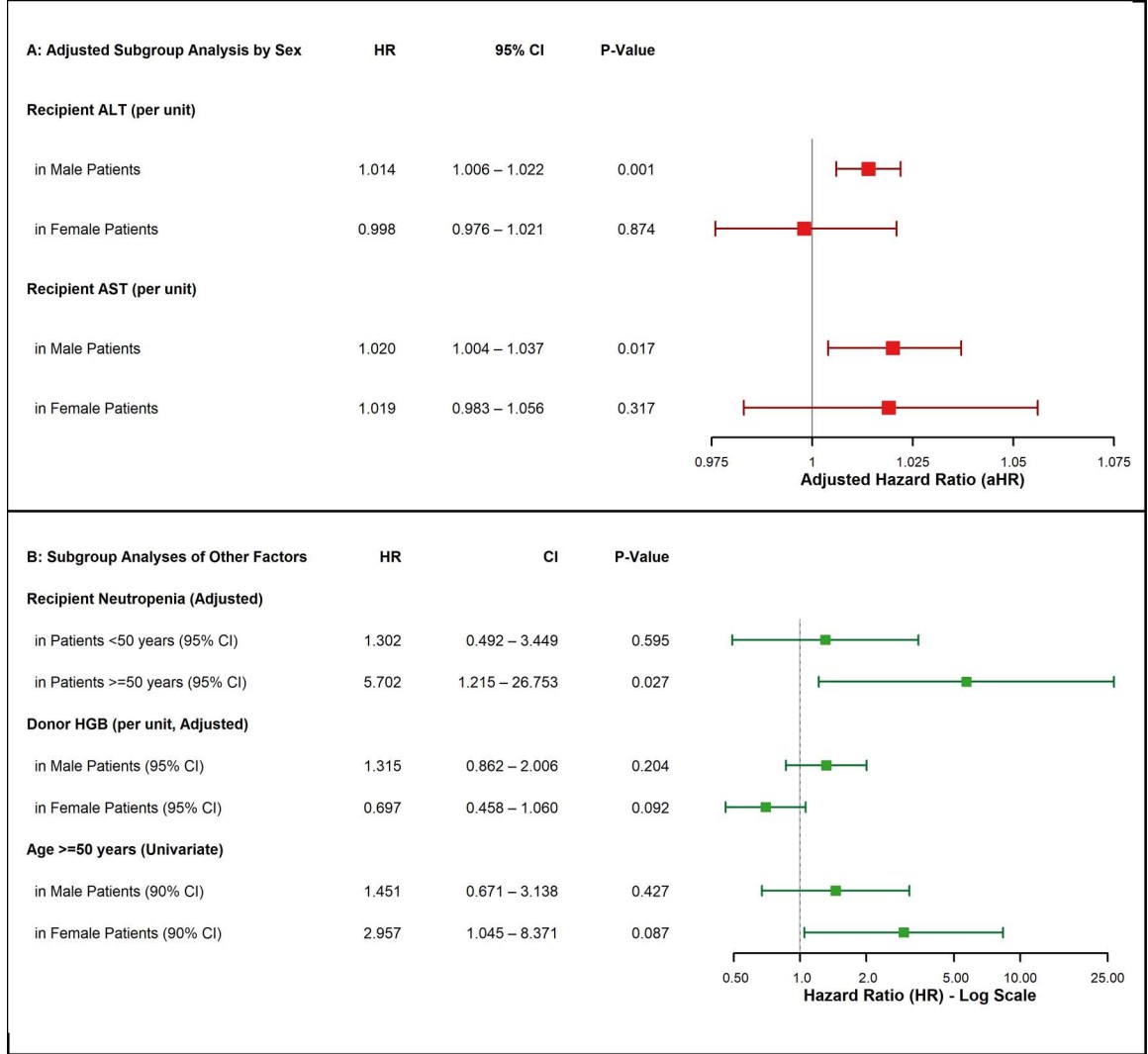

**Fig 5. Forest plots of exploratory subgroup analyses for key prognostic factors. (A) Adjusted Subgroup Analysis of Liver Enzymes by Sex.** The plot shows adjusted hazard ratios (aHRs) for recipient Alanine Aminotransferase (ALT) and Aspartate Aminotransferase (AST), stratified by patient sex. Models were adjusted for recipient age. Note the significant association with mortality risk is present only in male patients. **(B) Subgroup Analyses of Other Key Factors.** The plot shows hazard ratios (HRs) for Recipient Neutropenia (stratified by age < 50 vs. ≥ 50 years) and Donor Hemoglobin (stratified by patient sex). Analyses for Recipient Neutropenia and Donor Hemoglobin are from multivariate models adjusted for age and gender. *Abbreviations: aHR, adjusted Hazard Ratio; HR, Hazard Ratio; CI, Confidence Interval; ALT, Alanine Aminotransferase; AST, Aspartate Aminotransferase.*

a graft primed with inflammatory cytokines or effector cells, thereby exacerbating post-transplant inflammatory sequelae [22]. A second, alternative hypothesis historically involves direct iron toxicity and reactive oxygen species [23,24]. However, it is crucial to note that our entire cohort utilized peripheral blood stem cell (PBSC) grafts, which contain a negligible red blood cell mass compared to traditional bone marrow harvest. Therefore, direct systemic iron loading strictly from the graft itself is highly unlikely. Instead, we postulate that a constitutionally high donor hemoglobin may act as a systemic surrogate marker for an underlying donor metabolic state—such as altered iron homeostasis or subclinical pro-inflammatory pathways—which could indirectly influence graft composition, immune cell polarization, and early post-transplant inflammatory sequelae [10,25].

However, these general hypotheses alone fail to explain the centerpiece of our analysis: the paradoxical "rescue effect" observed specifically in neutropenic recipients. For this, we propose a more specific, multi-layered mechanism grounded in the fundamental biology of hematopoietic stem cell (HSC) regulation by erythropoietin (Epo). We hypothesize that a donor with a constitutionally high hemoglobin level has a correspondingly low systemic Epo tone. Based on emerging biological evidence, the graft from such a donor would be 'doubly advantaged.' Seminal studies have established that high systemic Epo levels act as a master regulator that not only (i) *impedes the physical migration and bone marrow homing* of hematopoietic stem/progenitor cells (HSPCs), but also (ii) actively *reprograms their lineage fate by suppressing non-erythroid options*, including the critical myeloid pathway [26]. This concept is further supported by recent clinical trials and mechanistic models from studies recent demonstrating that elevated EPO signaling in the stem cell niche actively impairs the engraftment and early homing of infused CD34 + cells, whereas attenuating this pathway accelerates neutrophil recovery [27,28]. Our finding is perfectly consistent with an inverse of this dual mechanism. A graft originating from a low-Epo environment would be released from both Epo-mediated migratory inhibition and myeloid suppression. This would render its HSPCs intrinsically permissive for more rapid marrow homing and robust myeloid differentiation—a decisive advantage for a neutropenic recipient in the critical race against early infection.

Furthermore, a third, complementary mechanism could involve the intrinsic epigenetic state of the HSCs themselves. Recent work in chronic stress erythropoiesis models (β-thalassemia) has demonstrated that the HSC pool can become epigenetically 'primed' for the erythroid lineage through complex signaling involving TGFβ and autophagy [29]. While this inherent erythroid priming might seem counter-intuitive to a myeloid rescue, it suggests that HSCs from a donor in a state of chronic erythropoietic demand exist in a hyper-responsive or 'poised' state. It is plausible that when transplanted into the powerful inflammatory, cytokine-rich milieu of a neutropenic host, these poised stem cells are uniquely capable of rapidly responding to the overriding signals demanding urgent myeloid regeneration. Taken together, these interconnected mechanisms transform our statistical observation into a tangible, testable biological hypothesis rooted in established principles of hematopoiesis.

The primary theoretical implication of our research is its challenge to simplistic, linear risk-assessment paradigms in allo-HSCT. Our data, supported by recent biological findings [26,30], suggest that the interplay between the host environment and graft potential is a critical, underappreciated axis in determining transplant outcomes. The practical implication, though strictly reserved for the future, is the potential for more nuanced and personalized donor selection strategies. For a profoundly immunocompromised patient—a group known to suffer high mortality from poor graft function [17]—the ideal graft may not simply be the one with the fewest intrinsic risk factors, but rather the one with the highest intrinsic potential for rapid myeloid reconstitution. However, it must be unequivocally stated that these findings are strictly hypothesis-generating and should not, under any circumstances, be used to guide current clinical practice.

It is essential to interpret our findings within the context of several significant limitations, in accordance with the STROBE guidelines. First and foremost, our study's primary limitation is its modest sample size (n = 94) and the correspondingly limited number of events (n = 30). This constrains statistical power and elevates the risk of a Type I error. While our central interaction finding was statistically robust —persisting even after sensitivity analyses adjusting for primary diagnosis, conditioning regimen, and relapse status—it must be interpreted with significant caution until independently replicated.. Additionally, the absence of an external validation cohort is a key limitation; therefore, our findings remain strictly exploratory and must be validated in independent, larger databases.

Second, the retrospective, single-center design inherently limits generalizability. Findings from our unique regional cohort in Western Iran may not be applicable to other populations. Specifically, our region is characterized by a high prevalence of red blood cell disorders, such as thalassemia minor. Although our institutional protocol strictly prioritizes donors without any hematological conditions—utilizing them only as an absolute last resort when no alternative donor is available—the unique genetic background of our local population could still theoretically influence baseline donor hemoglobin distributions and limits direct generalizability to populations with different genetic backgrounds. Additionally, the notably

high prevalence of pre-transplant neutropenia (43.6%) in our cohort requires consideration. This high rate likely reflects a heavily pre-treated patient population, with underlying factors such as advanced refractory disease or repeated cycles of salvage chemotherapy acting as potential unmeasured confounders that could independently influence both neutropenia and survival. Furthermore, this design is susceptible to unmeasured confounding. For instance, our retrospective dataset lacked granular data on established pre-transplant comorbidity indices (such as HCT-CI) and baseline recipient performance status (e.g., ECOG or Karnofsky scores). The absence of these standard variables prevents us from fully adjusting for the baseline physiological frailty of the recipients, which is a known independent predictor of mortality. Additionally, we lacked data on pre-transplant inflammatory markers in donors, such as C-reactive protein (CRP), which has been established as a potent prognostic factor in *recipients* [19,20]. The absence of such data prevents us from dissecting whether high hemoglobin is a direct biological effector or merely a surrogate for donor inflammation.

Third, while we analyzed engraftment kinetics—demonstrating a uniform median time to engraftment of 10 days across all patient subgroups with no significant influence from donor hemoglobin, recipient neutropenia, or their interaction (p = 0.812)—our study lacks the granular data required to directly test other downstream aspects of our biological hypotheses. This uniform engraftment suggests that the observed survival differences are likely driven by post-engraftment events, such as qualitative immune reconstitution, GVL effects, or susceptibility to late infections, rather than initial homing or primary graft failure. Without systematic data on these post-engraftment variables or documented rates of early infection, our proposed mechanism remains a plausible but unproven hypothesis.

While significant, the limitations of this study delineate a clear and compelling roadmap for future research. The primary contribution of this work is to catalyze a new line of inquiry. We propose a two-stage approach:

1. Validation: The essential first step is the validation of our statistical finding. Analysis of large, multi-center registry databases (e.g., CIBMTR, EBMT) is required to ascertain whether the interaction between recipient neutropenia and donor hemoglobin is a robust and generalizable phenomenon.2. Mechanistic Elucidation: Following validation, prospective studies are warranted. Drawing on established links between inflammation, EPO signaling, and graft function [30], these studies must be designed to co-collect standardized donor hemoglobin levels along with donor EPO levels, inflammatory markers (e.g., CRP), detailed graft cellular composition, and meticulously track qualitative immune reconstitution and rates of early infectious complications.

Only through such a systematic, multi-pronged approach can we move this observation from a novel statistical signal to a potential clinically actionable biomarker, ultimately refining the art of donor selection for our most vulnerable patients.

## Conclusion

This single-center, hypothesis-generating study of 94 allo-HSCT recipients identified a statistically robust context-dependent interaction between recipient neutropenia and donor hemoglobin. While we propose a mechanistic framework involving EPO-mediated effects on HSPC behavior, this remains speculative without direct validation. The primary contribution of this work is to generate a testable hypothesis for multi-center validation studies that should prospectively collect donor EPO levels, graft composition, and engraftment kinetics. Only through such validation can this observation be considered for potential clinical application in future donor selection algorithms.

## Supporting information

**S1 Table. Sensitivity analysis comparing multiple imputation (MI) and complete case analysis (CCA) results for uric acid and ALP (adjusted for age and gender).**
(DOCX)

**S1 Checklist. STROBE Statement—Checklist of items for a cohort study.**
(DOCX)

**S1 Dataset. Minimal anonymized underlying dataset.** This file contains the de-identified patient data used for the primary survival and interaction analyses.
(XLSX)

## Acknowledgments

The authors would like to thank the staff of the Kermanshah Bone Marrow Transplantation Centre for their assistance in the course of this study. Furthermore, we extend our sincere gratitude to the Aamar Afzar Institute, and particularly Dr. Abbas Keshtkar, for their invaluable training courses on statistics and research methodology in medical and humanities sciences.

## Author contributions

**Conceptualization:** Mohammadreza Eslami, Mahdi Mehrabi, Mehrdad Payandeh.

**Data curation:** Mohammadreza Eslami, Mahdi Mehrabi.

**Formal analysis:** Mohammadreza Eslami.

**Methodology:** Mohammadreza Eslami, Mahdi Mehrabi.

**Project administration:** Mehrdad Payandeh.

**Supervision:** Mehrdad Payandeh.

**Writing – original draft:** Mohammadreza Eslami, Mahdi Mehrabi.

**Writing – review & editing:** Mehrdad Payandeh.

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
