## [Decision Letter · Decision Letter 0]

11 Mar 2026

PONE-D-25-61939High donor hemoglobin interacts with pre-transplant recipient neutropenia to modulate mortality after allogeneic hematopoietic stem cell transplantation: a single-center, retrospective, real-world studyPLOS One

Dear Dr. Eslami,

Thank you for submitting your manuscript to PLOS ONE. After careful consideration, we feel that it has merit but does not fully meet PLOS ONE’s publication criteria as it currently stands. Therefore, we invite you to submit a revised version of the manuscript that addresses the points raised during the review process.

A letter that responds to each point raised by the academic editor and reviewer(s). You should upload this letter as a separate file labeled ’Response to Reviewers’.A marked-up copy of your manuscript that highlights changes made to the original version. You should upload this as a separate file labeled ’Revised Manuscript with Track Changes’.An unmarked version of your revised paper without tracked changes. You should upload this as a separate file labeled ’Manuscript’.

We look forward to receiving your revised manuscript.

Kind regards,

Daniel Thomas, MD

Academic Editor

PLOS One

Journal Requirements:

1.Please ensure that your manuscript meets PLOS ONE’s style requirements, including those for file naming. The PLOS ONE style templates can be found at

a) If there are ethical or legal restrictions on sharing a de-identified data set, please explain them in detail (e.g., data contain potentially identifying or sensitive patient information, data are owned by a third-party organization, etc.) and who has imposed them (e.g., a Research Ethics Committee or Institutional Review Board, etc.). Please also provide contact information for a data access committee, ethics committee, or other institutional body to which data requests may be sent

3. Please note that your Data Availability Statement is currently missing [the repository name and/or the DOI/accession number of each dataset OR a direct link to access each database]. If your manuscript is accepted for publication, you will be asked to provide these details on a very short timeline. We therefore suggest that you provide this information now, though we will not hold up the peer review process if you are unable.

Additional Editor Comments:

Eslami et al present a retrospective association study showing donor hemoglobin has an effect on overall survival in allogeneic stem cell transplantation and claim this can mitigate the effect of recipient neutropenia. The findings would be of interest to the broader blood cancer community but would require validation in other cohorts. A limitation is the relatively small sample size and relative low effect size in terms of hazard ratio. Also there is little discussion of other variables that independently correlate with poor survival in donors vs recipients. The references are not exhaustive nor current.

1] Can the authors perform analyses with standard prognostic factors that predict outcome in allogeneic transplantation?

2] Why did the authors choose to examine donor hemoglobin?

3] Can data from another published cohort where donor hemoglobin levels are available for validation?

Reviewer’s Responses to Questions

**Comments to the Author**

1. Is the manuscript technically sound, and do the data support the conclusions?

Reviewer #1: No

2. Has the statistical analysis been performed appropriately and rigorously? 

Reviewer #1: Yes

3. Have the authors made all data underlying the findings in their manuscript fully available?

Reviewer #1: Yes

4. Is the manuscript presented in an intelligible fashion and written in standard English?

Reviewer #1: Yes

5. Review Comments to the Author

Reviewer #1: This manuscript presents an interesting, hypothesis-generating exploration of potential host–graft interactions in alloHSCT. The conceptual framework is novel and could be of interest to the field. However, there are several methodological and reporting issues that currently limit the interpretability and robustness of the findings. Addressing the points below would substantially strengthen the scientific validity of the work and clarify how readers should interpret these exploratory associations.

Main considerations:

1) Study design and variable selection

• Several well-established alloHSCT prognostic variables are not included in the descriptive tables or regression models (e.g., primary diagnosis, disease status at transplant, donor type, stem cell source, GVHD prophylaxis, comorbidity indices such as HCT-CI, and performance status). Given the strong known effects of these factors on outcomes, omission raises concern for residual confounding and omitted variable bias. Suggestion: consider expanding baseline characteristics and, if feasible, incorporating key transplant-specific covariates into univariate and multivariable analyses. If this is not possible due to data limitations, this should be clearly acknowledged and the conclusions further tempered.

• The rationale for selecting pre-alloHSCT recipient neutropenia and donor haemoglobin as candidate predictors would benefit from clearer justification in the Introduction, grounded in existing literature or biological plausibility.

2) Missing data and cohort construction

• Approximately 19% of patients were excluded due to missing data. The operational definition of “>30% incomplete medical records” is unclear, and it is not evident whether partial data recovery was attempted. Suggestion: please clarify how missingness was defined and whether attempts were made to retrieve incomplete records. A brief discussion of potential selection bias introduced by these exclusions would strengthen transparency.

3) Outcomes and mechanistic framing

• Overall survival is used as the primary outcome, yet much of the biological interpretation centres on graft function and host–graft interaction. Key transplant outcomes that would directly inform these hypotheses (e.g., time to neutrophil/platelet engraftment, primary/secondary graft failure, pure red cell aplasia) are not presented. Suggestion: if available, reporting engraftment kinetics and graft-related outcomes would substantially strengthen the biological plausibility of the proposed mechanisms. If unavailable, this limitation should be emphasized when interpreting the findings.

• The high prevalence of pre-alloHSCT neutropenia (43.6%) would benefit from contextualization (e.g., disease status, recent therapy, incomplete count recovery, disease biology).

4) Transplant-specific and biological considerations

• The hypothesis that donor haemoglobin reflects graft “vigour” is intriguing but difficult to evaluate without key transplant variables (stem cell source, cell dose, graft manipulation, GVHD prophylaxis, ABO matching) and engraftment data.

• The discussion of iron toxicity and red cell mass would benefit from:

- Reporting graft source (PBSC vs BM vs CBU), given typical low red cell content in PBSC grafts

- Information on donor/recipient iron studies and baseline iron overload

- Consideration of ABO matching and its relevance to red cell-related complications

• Recipient neutropenia is likely influenced by disease biology, remission status, and prior therapy (e.g., HMA exposure), which are not explored and may confound observed associations. These factors warrant further discussion.

•Given the single-centre design, contextual information on the local population (e.g., prevalence of red cell disorders in the donor pool) may be relevant to interpreting donor haemoglobin distributions and generalizability.

5) Statistical interpretation and scope of inference

• The manuscript occasionally uses language suggesting confirmation or mitigation of effects. Given the retrospective, exploratory design, it would be more appropriate to frame these findings strictly as associations requiring external validation.

• Multiple exploratory comparisons are performed. Suggestion: please explicitly discuss the risk of false-positive findings and how readers should interpret statistical significance in this hypothesis-generating context.

• Kaplan–Meier and predicted survival figures should include corresponding p-values and clearer annotation to aid interpretation.

Additional Considerations:

• Please report the study timeframe with precise start and end dates.

• Consider consolidating overlapping Methods subsections for clarity.

• Table 1 would benefit from clearer categorization of disease indications and disease status at transplant.

• Given overall survival is the primary endpoint, please report median OS and time-specific survival estimates.

The conceptual idea is interesting and potentially hypothesis-generating for future studies. Strengthening variable reporting, addressing missing data more transparently, incorporating key alloHSCT-specific confounders (where feasible), and tempering causal or confirmatory language would substantially improve the manuscript and help readers appropriately interpret the findings.

6. PLOS authors have the option to publish the peer review history of their article (what does this mean?). If published, this will include your full peer review and any attached files.

Reviewer #1: No

---

## [Author Response · Author response to Decision Letter 1]

29 Apr 2026

Dear Dr. Daniel Thomas,

Academic Editor, PLOS ONE,

We would like to express our profound gratitude to you and the esteemed Reviewer for the thoughtful, rigorous, and highly constructive evaluation of our manuscript. Your insightful comments have been instrumental in refining our methodology, tempering our conclusions, and fundamentally improving the biological plausibility of our hypothesis.

Before addressing the scientific comments, we feel it is necessary to offer a brief, sincere context regarding the timing of our revision. Due to recent severe geopolitical conflicts and an unprecedented 47-day national internet blackout in Iran (until April 15), our research team was entirely cut off from the journal’s editorial system. We only very recently regained access to view the reviewers’ valuable reports. While we have managed to submit this revision well within the journal’s formal deadline, the drastically compressed timeframe forced us to work under extraordinary logistical and psychological constraints. We sincerely apologize for our prolonged silence and ask for your kind understanding, assuring you that we have spared no effort to maintain the highest standard of scientific rigor in this revision.

Prompted by the Reviewer’s excellent request to incorporate additional transplant-specific variables, we conducted a rigorous, manual re-audit of our entire dataset. This process allowed us to successfully retrieve crucial variables (such as exact stem cell source, ABO compatibility, and GVHD prophylaxis) and correct a minor data entry discrepancy regarding the continuous donor hemoglobin variable. Consequently, we have re-run all statistical models. Remarkably, this rigorous data verification refined and strengthened our core hypothesis: while the main effect of donor hemoglobin in the overall cohort is no longer strictly significant (p=0.148), the crucial interaction between recipient neutropenia and donor hemoglobin remains highly significant and robust (Interaction p=0.013). This perfectly aligns with our central premise that the prognostic impact of donor hemoglobin is not absolute, but deeply context-dependent.

We have addressed all concerns point-by-point below. Changes in the manuscript are highlighted in the ‘Revised Manuscript with Track Changes’ file.

Response to the Academic Editor

Editor Comment 1: Can the authors perform analyses with standard prognostic factors that predict outcome in allogeneic transplantation?

Response: We fully agree with this essential requirement. We have now extensively updated Table 1 (Pages 20-22) to include critical standard prognostic factors, including primary diagnosis categorization, disease status at transplant (relapsed vs. non-relapsed), HLA match status, stem cell source (which was uniquely 100% PBSC), ABO compatibility, exact conditioning regimens, and GVHD prophylaxis.

Furthermore, to address potential residual confounding by these standard variables without violating the Events-Per-Variable (EPV) constraints of our modest sample size, we performed targeted sequential sensitivity analyses. As detailed in the Methods (Page 5) and Results (Pages 8-9), we sequentially added key transplant covariates (primary diagnosis, conditioning regimen, relapse status) to our primary Gompertz model. The interaction between neutropenia and donor hemoglobin remained robust in all sensitivity models. Additionally, we explicitly acknowledged the lack of data on certain physiological frailty indices (such as HCT-CI and ECOG performance status) as a limitation of our retrospective design (Page 13).

Editor Comment 2: Why did the authors choose to examine donor hemoglobin?

Response: We appreciate the opportunity to clarify our rationale. While routine donor parameters like baseline hemoglobin are checked strictly for donor safety, they are rarely investigated as predictors of recipient outcomes. Biologically, higher donor hemoglobin within the physiological range may act as a surrogate marker for robust hematopoietic reserve—representing an inherently vigorous graft (the "seed"). Conversely, severe recipient neutropenia reflects a depleted, heavily pre-treated marrow microenvironment and profound host immunosuppression (the "soil"). We hypothesized that the impact of this highly vulnerable "soil" could be uniquely modified by the physiological vigor of the "seed" beyond standard HLA matching. We have explicitly added this "soil and seed" conceptual framework to the Introduction (Page 3) to better justify our variable selection.

Editor Comment 3: Can data from another published cohort where donor hemoglobin levels are available for validation?

Response: We completely agree that external validation is the gold standard for any novel biomarker association. However, after extensive searching and literature review, we found that major registry databases (such as CIBMTR or EBMT) and previously published retrospective cohorts do not routinely collect or publicly share granular donor physiological data (like continuous baseline hemoglobin or donor EPO levels) paired with recipient pre-transplant immune status. Extracting this specific matched data requires a prospective, multicenter design or a dedicated registry-level proposal, which is beyond the scope of the current single-center study. Therefore, we have strictly framed our study as "hypothesis-generating" and explicitly stated in the Limitations and Conclusion sections (Pages 12-14) that these findings remain exploratory and must not guide clinical practice until prospectively validated.

Editor Comment 4: The references are not exhaustive nor current.

Response: We thank the Editor for this observation. We have carefully reviewed our bibliography and enriched our discussion by integrating multiple recent and highly relevant peer-reviewed studies from recent five years. These newly added citations specifically address recent advancements in non-HLA donor characteristics, the prognostic implications of poor graft function, contemporary NMDP/CIBMTR guidelines, and novel mechanistic models demonstrating how elevated EPO signaling directly impairs stem cell engraftment kinetics.

Response to Reviewer #1

Reviewer Point 1: Study design and variable selection

Several well-established alloHSCT prognostic variables are not included... The rationale for selecting pre-alloHSCT recipient neutropenia and donor haemoglobin as candidate predictors would benefit from clearer justification.

Response: We are highly grateful for this critical observation.

1. Variables: We have comprehensively updated Table 1 to include Stem Cell Source, GVHD Prophylaxis, detailed Conditioning Regimens, ABO match, and Relapse Status. These variables were also tested in sensitivity models (Pages 8-9). Regarding HCT-CI and performance status, we unfortunately lacked granular registry data for these specific scores. We have transparently declared this unmeasured confounding in the Limitations section (Page 13).

2. Rationale: We have significantly rewritten the Introduction (Page 3) to articulate the "soil and seed" framework, providing a clear biological justification for exploring the interaction between host immunocompromise and graft vigor.

Reviewer Point 2: Missing data and cohort construction

Approximately 19% of patients were excluded due to missing data. The operational definition of “>30% incomplete medical records” is unclear...

Response: We apologize for the ambiguity. We have clarified in the Methods section (Page 4) that the exclusion threshold was strictly operationalized as missing data across more than 30% of the predefined core baseline and transplant-specific variables, despite rigorous retrospective attempts to retrieve this missing information from hospital archives. We agree that this exclusion could introduce potential selection bias, which we attempted to mitigate by employing Multiple Imputation for the remaining missing covariates.

Reviewer Point 3: Outcomes and mechanistic framing

Key transplant outcomes that would directly inform these hypotheses (e.g., time to neutrophil/platelet engraftment) are not presented. The high prevalence of pre-alloHSCT neutropenia (43.6%) would benefit from contextualization.

Response: This is a superb point.

1. Engraftment Kinetics: We have extracted the relevant data and added a new subsection "Engraftment kinetics" in the Results (Page 8). The median time to neutrophil engraftment was uniformly 10 days across all patient subgroups, with no significant differences influenced by neutropenia or donor hemoglobin. We have added to the Discussion (Page 13) that this uniform early engraftment suggests the observed survival differences are likely driven by post-engraftment events (e.g., qualitative immune reconstitution, GvL, or susceptibility to late infections) rather than initial homing or primary graft failure.

2. Neutropenia Context: We have expanded the Discussion (Page 12) to contextualize the high prevalence of pre-transplant neutropenia. We clarified that this reflects our specific regional cohort’s nature: heavily pre-treated patients with advanced refractory disease or those who underwent repeated cycles of salvage chemotherapy.

Reviewer Point 4: Transplant-specific and biological considerations

The discussion of iron toxicity and red cell mass would benefit from reporting graft source... given typical low red cell content in PBSC grafts.

Response: We sincerely thank the Reviewer for pointing out this critical biological nuance; it has vastly improved the manuscript. We have now reported in Table 1 that 100% of our cohort uniquely received PBSC grafts. Because PBSC grafts contain a negligible red blood cell mass compared to bone marrow harvests, direct systemic iron loading from the graft itself is highly unlikely. We have completely rewritten this section of the Discussion (Page 10) to reflect this reality. We now postulate that rather than direct iron toxicity, a constitutionally high donor hemoglobin may act as a systemic surrogate marker for an underlying donor metabolic state (e.g., altered iron homeostasis or subclinical pro-inflammatory pathways) that indirectly influences graft composition. Furthermore, we added a discussion regarding the high prevalence of red blood cell disorders (e.g., thalassemia minor) in our specific regional population in Western Iran, acknowledging how this unique genetic background might influence baseline hemoglobin distributions and limit direct generalizability (Page 12).

Reviewer Point 5: Statistical interpretation and scope of inference

The manuscript occasionally uses language suggesting confirmation or mitigation of effects. Please explicitly discuss the risk of false-positive findings. Kaplan–Meier and predicted survival figures should include corresponding p-values.

Response: We fully agree and have meticulously revised the manuscript’s tone throughout. We have replaced causal language (e.g., "mitigates," "confirms") with associative and cautious language (e.g., "associated with a trend," "potentially protective"). We have heavily emphasized the exploratory, hypothesis-generating nature of the study, the constraints of the modest sample size, the risk of unmeasured confounding, and the absolute necessity for external validation (Pages 11-13). P-values and risk tables are explicitly verified for all updated figures.

Additional Considerations:

Please report the study timeframe... Table 1 would benefit from clearer categorization... please report median OS and time-specific survival estimates.

Response:

• The precise study timeframe (September 5, 2015, to March 12, 2024) has been added to the Methods (Page 4).

• Table 1 has been extensively reorganized for clarity, explicitly separating recipient demographics, clinical characteristics, and transplant parameters.

• We have added the requested time-specific survival estimates to the Results (Page 6). Due to favorable long-term outcomes in our specific cohort, the median overall survival was not reached. The estimated 1-year, 3-year, and 5-year OS rates were 71.3%, 67.3%, and 67.3%, respectively.

We thank you once again for your time, patience, and expert guidance. We believe the manuscript is now substantially more robust, transparent, and biologically sound. We hope it is now deemed suitable for publication in PLOS ONE.

Sincerely,

Mohammadreza Eslami

---

## [Editor Report · Decision Letter 1]

4 May 2026

High donor hemoglobin interacts with pre-transplant recipient neutropenia to modulate mortality after allogeneic hematopoietic stem cell transplantation: an exploratory, single-center, retrospective, real-world study

PONE-D-25-61939R1

Dear Dr. Eslami,

We’re pleased to inform you that your manuscript has been judged scientifically suitable for publication and will be formally accepted for publication once it meets all outstanding technical requirements.

An invoice will be generated when your article is formally accepted. Please note, if your institution has a publishing partnership with PLOS and your article meets the relevant criteria, all or part of your publication costs will be covered. Please make sure your user information is up-to-date by logging into Editorial Manager at Editorial Manager® and clicking the ‘Update My Information’ link at the top of the page. For questions related to billing, please contact billing support.

Kind regards,

Daniel Thomas, MD

Academic Editor

PLOS One

Additional Editor Comments (optional):

The authors have significantly decreased their claims. This study is exploratory and hypothesis generating rather than conclusive. This is now indicated in title and abstract. All comments have been addressed adequately

Reviewers’ comments:

---

## [Editor Report · Acceptance letter]

PONE-D-25-61939R1

PLOS One

Dear Dr. Eslami,

I’m pleased to inform you that your manuscript has been deemed suitable for publication in PLOS One. Congratulations! Your manuscript is now being handed over to our production team.

Lastly, if your institution or institutions have a press office, please let them know about your upcoming paper now to help maximize its impact. If they’ll be preparing press materials, please inform our press team within the next 48 hours. Your manuscript will remain under strict press embargo until 2 pm Eastern Time on the date of publication. For more information, please contact onepress@plos.org.

Kind regards,

on behalf of

Dr. Daniel Thomas

Academic Editor

PLOS One